

# Universal properties of anisotropic dipolar bosons in two dimensions

**Juan Sánchez-Baena[1,3], Luis Aldemar Peña Ardila[2],**
**Gregory E. Astrakharchik[3] and Ferran Mazzanti[3]**

**1** Center for Complex Quantum Systems, Department of Physics and Astronomy,
Aarhus University, DK-8000 Aarhus C, Denmark
**2** Institut für Theoretische Physik, Leibniz Universität Hannover, Germany
**3** Departament de Física, Campus Nord B4-B5, Universitat Politècnica de Catalunya,
E-08034 Barcelona, Spain

## Abstract

The energy of ultra-dilute quantum many-body systems is known to exhibit a universal dependence on the gas parameter $x = na_0^d$, with $n$ the density, $d$ the dimensionality of the space ($d = 1, 2, 3$) and $a_0$ the $s$-wave scattering length. The universal regime typically extends up to $x \approx 0.001$, while at larger values specific details of the interaction start to be relevant and different model potentials lead to different results. Dipolar systems are peculiar in this regard since the anisotropy of the interaction makes $a_0$ depend on the polarization angle $\alpha$, so different combinations of n and $\alpha$ can lead to the same value of the gas parameter x. In this work we analyze the scaling properties of dipolar bosons in two dimensions as a function of the density and polarization dependent scattering length up to very large gas parameter values. Using Quantum Monte Carlo (QMC) methods we study the energy and the main structural and coherence properties of the ground state of a gas of dipolar bosons by varying the density and scattering length for a fixed gas parameter. We find that the dipolar interaction shows relevant scaling laws up to unusually large values of $x$ that hold almost to the boundaries in the phase diagram where a transition to a stripe phase takes place.

Ultra-dilute systems have recently gained renewed interest since the existence of liquid-like droplets of Bose mixtures was predicted [1], resulting in equilibrium densities orders of magnitude lower than what is found in other systems such as Helium [2,3]. In the context of Bose-Bose mixtures, the formation of this liquid state results from the delicate balance between the overall attractive mean-field energy arising from the competition between interspecies attraction and intraspecies repulsion, and a repulsive contribution caused by quantum fluctuations, which stabilizes the system. Bose-Bose self-bound droplets have been both described theoretically [1,4,5] and observed experimentally [6,7]

Ultradilute droplets have also been achieved in single-component dipolar systems. They result from the competition of the repulsion induced by a contact interaction term, and the dipole-dipole interaction (DDI) [8,9]. Dipolar droplets have also been predicted recently in

dipolar mixtures [10, 11]. More complex systems featuring spin-orbit interactions have also been reported to be able to form ultralow density droplets that can even show a striped pattern [12].

In general, quantum systems at zero temperature and very low densities are known to follow universal equations of state [13–18]. In all cases, the leading terms are given by the mean-field (MF) prediction, where the energy per particle $\epsilon(x)$ is a function of the gas parameter $x = na_0^d$, with $n$ the density, $d$ the dimensionality of the space, and $a_0$ the $s$-wave scattering length of the interatomic interaction. Within the lowest order Born approximation, the only relevant parameter in a pseudo-potential expansion of the interaction is $a_0$, while additional quantities like the s-wave effective range or other parameters from higher order partial waves do not contribute significantly. In the common case of central interactions, the scaling in $x$ starts to break down as the gas parameter exceeds a critical value $x_c \approx 0.001$. Below that value, all interaction sharing the same $s$-wave scattering length follow the MF + Beyond Mean Field prediction.

Within mean-field theory, the energy per particle is linear with the density, $\epsilon = gn/2$, with the coefficient of proportionality given by the coupling constant $g$, which defines the strength of the short-range interaction between bosons modeled by a pseudopotential, $V_{SR}(\mathbf{r}) = g\delta(\mathbf{r})$. In three dimensions, the relation between the coupling constant and the $s$-wave scattering length is linear, $g_{3D} = 4\pi\hbar^2 a_0/m$, where $m$ is the particle mass, resulting in a linear dependence of $\epsilon$ on the gas parameter $x$, $\epsilon/(\hbar^2/ma_0^2) = 2\pi x$. The situation is significantly more complicated in the two-dimensional case where $\epsilon$ is fully defined by the density $n$, as experimentally shown for 2D Bose gases in Refs. [19, 20].

In two dimensions, the dependence of the system properties on the $s$-wave scattering length $a_0$ occurs due to a quantum mechanical symmetry breaking. This is known as a quantum anomaly [21, 22]. In the context of ultracold gases with short range interactions, this quantum anomaly manifests itself through the symmetry breaking of the scale invariance present in the classical field treatment of the problem. Therefore, the quantum anomaly phenomenon generates deviations from the predictions established from the classical field results. Among these, the modification of the frequency of the breathing mode in two dimensions has been both predicted theoretically for Bosons [23, 24] and Fermions [22, 25] and observed experimentally in fermionic systems [26, 27]. Also, the change in the power-law exponents associated with long-range phase correlations in the system has been recently observed [28]. Still the dependence of the coupling constant $g_{2D}$ on $a_0$ is extremely weak and comes through the logarithm of the gas parameter, $g_{2D} = 4\hbar^2/(m|\ln x|)$ [15, 29], with further perturbative terms introducing recursive contributions of the form $\ln|\ln x|$ [30–35]. A similar perturbative structure appears in the thermal effects associated with the Berezinskii-Kosterlitz-Thouless (BKT) phase transition [36, 37]. However, in the latter case, the recursive term $\ln|\ln x|$ does not play a major role in typical experimental conditions, since its contribution is always smaller than that of a dimensionless experimental parameter [36]. In fact, the weak dependence of the beyond mean field corrective terms limits the validity of the MF theory to exponentially small values of $x$. Indeed, it was shown in Ref. [38] that it is necessary to reach values as small as $x \sim 10^{-100}$ to see that the influence of the beyond mean field corrections is in general negligible. As long as the energy is concerned, though, a cancellation of the logarithmic corrective terms leads to very small deviations from the MF prediction below $x \sim 10^{-3}$.

The inclusion of dipolar physics brings a whole new degree of theoretical sophistication to a proper description of the equation of state $\epsilon$. This is because, while technically speaking the dipolar interaction is short-ranged, its extension is large compared to other typical short-range potentials. On top of that, the dipole-dipole interaction (DDI) depends not only on the distance but also on the relative orientation of the constituents, introducing additional degrees of freedom in the Hamiltonian. In the specific case of polarized two-dimensional dipoles, the

energy per particle was shown in Ref. [39] to follow the universal prediction up to the critical value $x_c$. Due to the anisotropy of the interaction, however, the $s$-wave scattering length of polarized dipoles in 2D depends on the polarization angle $\alpha$ as $a_0(\alpha)/a_d \approx e^{2\gamma}(1 - 3\lambda^2/2)$, with $\lambda = \sin(\alpha)$, $\gamma = 0.577\ldots$ the Euler's constant, and $a_d = mC_{dd}/4\pi\hbar^2$ the dipolar unit of length [39]. The DDI potential

$$V_{dd}(\mathbf{r}) = \frac{C_{dd}}{4\pi}\left[\frac{1 - 3\lambda^2\cos^2\theta}{r^3}\right],\tag{1}$$

describes dipoles moving in the XY plane while being oriented externally by a polarization field contained in the XZ plane. It is important to remark that increasing the tilting angle beyond a critical value $\alpha_c \simeq 0.615$ makes the system collapse. This is because for $\alpha > \alpha_c$ the DDI becomes negative for values of $\theta$ around zero, while the quantum pressure is not enough to overcome it.

The dependence of $a_0$ on the polarization angle implies that the same value of the gas parameter $x = na_0^2$ can be achieved in many different ways by properly adjusting $n$ and $\alpha$. In particular, increasing $\alpha$ leads to a reduction of the repulsion of the DDI, thus implying that $n$ must be increased to keep $x$ constant. In Ref. [39] the authors showed that different combinations leading to the same $x$ yield the same $\epsilon(x)$, even for low values of $x$ that are larger than $x_c$. In this way, the equation of state of bosonic dipoles in two dimensions seem to follow a universal dipolar curve. Universal properties of dipolar bosons in two dimensions have also been discussed in Ref. [40].

In this work we explore this universality among dipoles featuring different orientations up to the large gas parameter values way above $x_c$. We aim to characterize the degree of universality not only in the energy per particle, but also in other observables directly related to the structure of the system and its coherence properties. This is done by performing diffusion Monte Carlo (DMC) simulations of $N$ dipolar bosons moving in a box with periodic boundary conditions contained in the XY plane. The system is described by the many-body Hamiltonian

$$H = -\frac{\hbar^2}{2m}\sum_{j=1}^{N}\nabla_j^2 + \sum_{i<j}V_{dd}(\mathbf{r}_{ij}),\tag{2}$$

with $m$ the mass and $\mathbf{r}_{ij}$ the relative position vector. While DMC produces statistically exact energies, its convergence properties benefit from the use of a variational guiding wave function $\Phi_0(\mathbf{r}_1, \mathbf{r}_2, \ldots, \mathbf{r}_N)$ to drive the dynamics in imaginary time. In this work we build $\Psi_0$ as a Jastrow pair-product form

$$\Phi_0(\mathbf{r}_1, \mathbf{r}_2, \ldots, \mathbf{r}_N) = \prod_{i<j}f(\mathbf{r}_{ij}),\tag{3}$$

with $f(\mathbf{r})$ the solution of the zero-energy two body problem, matched with a phononic tail at a distance $r_m$ that is variationally optimized, as described in [39]. As we wish to compare different dipolar systems that have the same gas parameter $x$ and different tilting angles, simulations are performed for different values of the density $n$ as $\alpha$ changes such that $x$ remains constant. We find that $N = 100$ particles are enough for all the gas parameter values explored but $x = 350$, where $N = 200$ has been used. We have checked that our results remain essentially unchanged when keeping $n$ constant while increasing $N$ and the box size $L$. We further elaborate on their dependency on the finite size of the system below.

We start the discussion of the results by first addressing the energy per particle of the system, since this is the driving quantity characterizing universality among different quantum systems at zero temperature. In these systems, universality takes place when the energy per particle, expressed in scattering length units, becomes a function of the gas parameter $x = na_0^2$

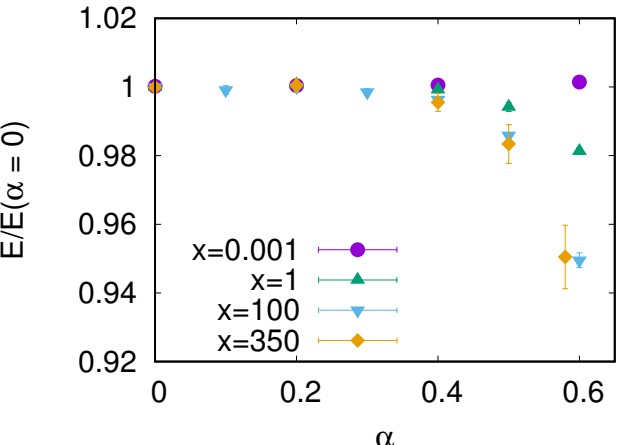

Figure 1: Ratio of the bulk DMC energy per particle, in units of the scattering length $a_0(\alpha)$ to $E(\alpha = 0, x)$, for different values of the gas parameter $x$. The maximum tilting angle used for $x = 350$ is $\alpha = 0.58$.

only. In the present case where we compare the same (dipolar) interaction at different polarization angles $\alpha$, universality implies that all ratios $E(\alpha)/E_0(\alpha)$ with $E_0(\alpha) = \hbar^2/ma_0^2(\alpha)$, must collapse to the same curve $\epsilon(x)$. Figure 1 shows the bulk DMC energies per particle in dimensionless form for several values of the gas parameter $x$ and polarization angle $\alpha$. We report the ratio of the energy (in units of $E_0(\alpha) = \hbar^2/ma_0^2(\alpha)$) to the energy at $\alpha = 0$ (in units $E_0(0) = \hbar^2/ma_0^2(0)$), so that all curves in the figure start at one. Notice that the maximum tilting angle explored for $x = 350$ is $\alpha = 0.58$, as for larger values the ground state of the system lays in the stripe phase [41]. A perfect universal behaviour would correspond to $E/E(\alpha = 0) = 1$ for all polarization angles where the system is still in the gas phase. Surprisingly, and as it can be observed from the figure, the universal behavior holds for all $\alpha \lesssim 0.4$, while at larger angles slight deviations less or equal than 5% can only be seen at anomalously large values of $x \gtrsim 100$, which lays orders of magnitude above $x_c$. In this sense, the energy for any value of $\alpha$ can be well approximated with an error no larger than 5% when its value at any other single $\alpha$ (for instance $\alpha = 0$) is known. This scaling property allows for the computation of a single curve that can be property rescaled and used as an input to alternative mean-field models. We have found that a good fit to the DMC energies is given by the expression

$$\epsilon_{\text{full}} = E/N = \left( \frac{\hbar^2}{ma_0^2(\alpha)} \right) \exp\left[ A(\ln(x) + C)^l + B \right], \tag{4}$$

where $A = 0.920, B = -7.917, C = 8.0$, and $l = 1.117$ Furthermore and in agreement with Ref [39], in the universality regime of gas parameter values $x \lesssim 10^{-3}$, the energy of the dipolar gas can also be well approximated by the mean-field prediction

$$\epsilon_{\text{MF}} = E_{\text{MF}}/N = \left( \frac{\hbar^2}{2ma_0^2(\alpha)} \right) \frac{4\pi x}{|\ln(x)|}. \tag{5}$$

We show in Fig. (2) a comparison between the DMC energies and the values obtained using the expressions in Eqs. (4) and (5). As it can be seen from the Figure, the mean-field prediction of Eq. (5) closely reproduces the DMC energies for $x \lesssim 10^{-3}$, fairly close to the limit of validity of the universal equation of state. Beyond that point, the mean field functional drastically deviates from the DMC energies as well as from the prediction of Eq. (4).

It is also interesting to discuss the behavior of the energy expressed in dipole units so that the energy scale is set to $E_0 = \hbar^2/ma_d^2$ for all polarization angles. Since the scattering length

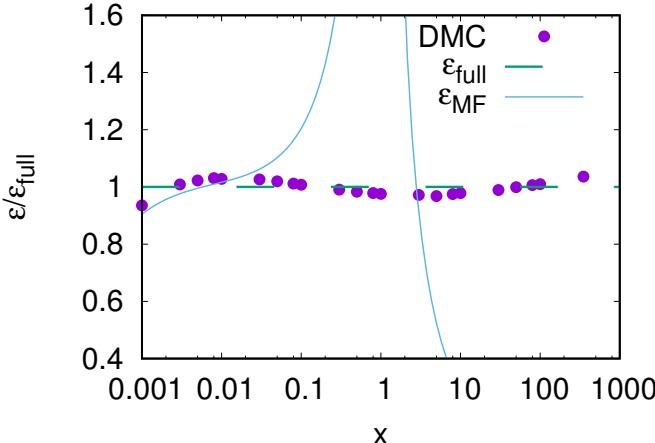

Figure 2: DMC energies per particle (symbols), $\epsilon_{\text{full}}$ (dashed line) and $\epsilon_{\text{MF}}$ (solid line), all in scattering length units, divided by $\epsilon_{\text{full}}$ as a function of the gas parameter. Here, $\epsilon$ denotes an energy per particle.

decreases with increasing $\alpha$ due to the anisotropy of the DDI, one readily notices from the curves in Fig. 1 that the energy increases for increasing $\alpha$, at least for $\alpha \lesssim 0.4$. This may seem to be a counteractive effect, as by increasing $\alpha$ the interaction becomes less repulsive almost everywhere. However, the density has to be increased when the scattering length is reduced in order to keep the gas parameter constant. In this way, the net increase in the energy is the result of two competing effects.

Next we discuss the structural properties of the system, starting with the pair distribution function $g(\mathbf{r})$, which is defined as

$$g(\mathbf{r} = \mathbf{r}_{12}) = \frac{N(N-1)}{nN} \frac{\int \mathbf{dr}_3 \cdots d\mathbf{r}_N \, |\Psi(\mathbf{r}_1, \mathbf{r}_2, \cdots, \mathbf{r}_N)|^2}{\int \mathbf{dr}_1 \cdots d\mathbf{r}_N \, |\Psi(\mathbf{r}_1, \mathbf{r}_2, \cdots, \mathbf{r}_N)|^2} \, . \tag{6}$$

This quantity measures the probability to find two particles at a relative distance given by the position vector $\mathbf{r}$. Considering the anisotropy present in the system, it is convenient to perform a partial waves expansion of $g(\mathbf{r})$ in the form

$$g(\mathbf{r}) = \sum_{m=0}^{\infty} g_{2m}(r) \cos(2m\theta),$$

with $(r, \theta)$ the polar coordinates. Due to the bosonic symmetry, only even order modes contribute to this expansion. In this way, the emergence of anisotropic effects in the structure of the system is manifested by the presence of non-vanishing $g_{2m}(r)$ terms with $m > 1$. In practice, though, we have found that higher order modes produce a negligible contribution when compared with the first two.

We focus on two main aspects concerning the pair distribution function: the effect of the anisotropy, and the possible scaling of $g(\mathbf{r})$ for different tilting angles $\alpha$. Results for $g_0(r)$ and $g_2(r)$ are shown in the left and right panels of Fig. (3) for increasing values of the gas parameter and polarization angle. In these plots all distances have been scaled by the corresponding scattering lengths, which is different for different values of $\alpha$. As it can be seen from the left upper and middle panels, for $x = 0.001$ and $x = 1$ the isotropic modes are equal, regardless of the value of $\alpha$. Similarly to the total energy discussed above, the pair correlation functions follow a universal trend, even for values of the gas parameter $x$ as large as 350, where deviations from a common curve are evident only at the largest polarization angle considered,

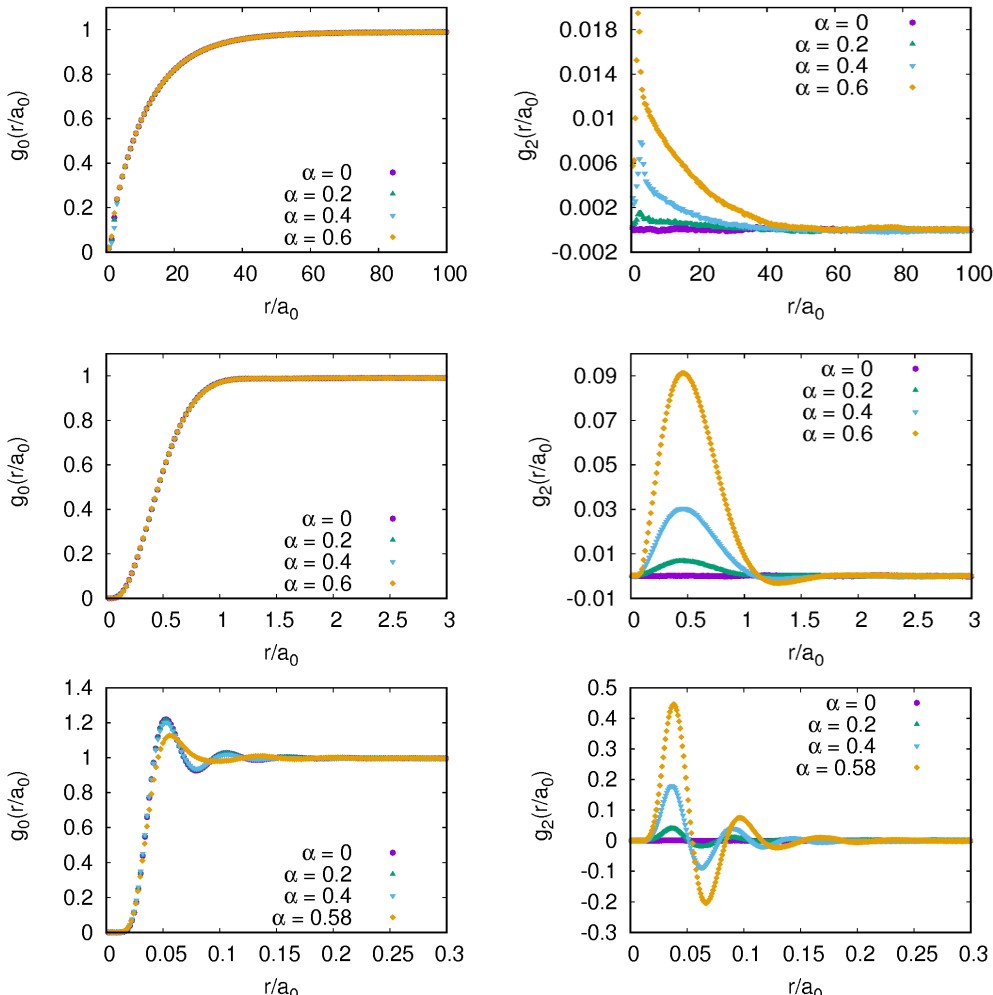

Figure 3: Isotropic (left plots) and first anisotropic modes (right plots) of the pair distribution function for $x = 0.001$ (top), $x = 1$ (middle) and $x = 350$ (bottom) and for different values of the tilting angle.

$\alpha = 0.58$. In this sense, the behavior of the isotropic mode of the pair distribution function shows a universal dipolar behavior that extends far beyond what is found in other quantum many-body systems [18].

The degree of anisotropy present can be measured by the strength of the $g_2(r)$ mode, which is depicted in the right panels of the same figure. As it can be seen, none of the curves are equal, not even at the lowest value of $x = 0.001$. This indicates that pure anisotropic effects in $g(\mathbf{r})$ do not scale, in contrast to what happens with the isotropic mode. In any case, it should be noticed that the relative strength of the anisotropic mode to the isotropic one is always small in the range of $x$ and $\alpha$ values considered, except for the largest ones. In this way one can conclude that the impact on the anisotropy of the interaction in the spatial structure of the system only affects significantly the dipolar gas close to the transition to the stripe phase.

From the pair distribution function one can obtain the static structure factor $S(\mathbf{k})$ by direct Fourier transform

$$S(\mathbf{k}) = 1 + n \int d\mathbf{r}\, e^{i\mathbf{k}\mathbf{r}}(g(\mathbf{r}) - 1)\,. \tag{7}$$

This quantity characterizes spatial ordering in the system, as periodic repetitions in space show up as peaks in $S(\mathbf{k})$. The dipolar system is known to enter the stripe phase at large densities

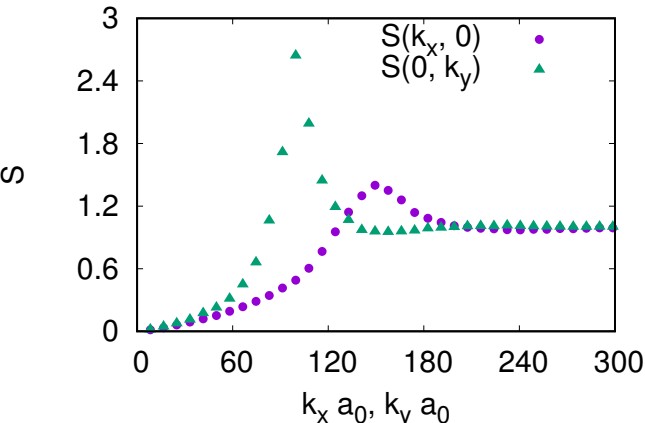

Figure 4: Static Structure factor $S(k_x, 0)$ and $S(0, k_y)$ computed for $x = 350$, $\alpha = 0.58$, with momenta scaled with the tilting-dependent scattering length $a_0(\alpha)$.

and polarization angles. In this respect, the $x = 350, \alpha = 0.58$ point lays very close to the transition line [41]. Even though this work is restricted to the study of dipolar gases, the gas parameters and tilting angles explored reach values large enough such that signs of spatial ordering along the direction of maximal repulsion of the interaction are visible. This is seen in Fig. (4), where we show $S(k_x, 0)$ and $S(0, k_y)$ for the largest values $x = 350$, $\alpha = 0.58$. This quantity has been obtained using the extrapolated estimator, which corrects to first order the bias caused by the trial wave function in the evaluation of expectation values of operators $\hat{O}$ that do not commute with the Hamiltonian. The extrapolated estimator is computed as the ratio $\langle \hat{O} \rangle_{\text{ext.}} = \langle \hat{O} \rangle_{\text{DMC}}^2 / \langle \hat{O} \rangle_{\text{VMC}}$, where the labels "VMC" and "DMC" stand for Variational and Diffusion Monte Carlo, respectively. As it can be seen, $S(0, k_y)$, which corresponds to the direction of maximum repulsion, shows a pronounced peak that is absent in $S(k_x, 0)$. This is the triggering sign of spatial ordering along the $Y$ direction, in what constitutes an anisotropic gas, a precursor of the supersolid stripe phase. Being S(**k**) the Fourier transform of $g(\mathbf{r})$, the scaling properties presented by the static structure factor in terms of $x$ and $\alpha$ are essentially the same ones presented by the pair distribution function analyzed above.

In order to clarify the origin of the universality in the energy as a function of the gas parameter, it is worth mentioning that the pair distribution function can be directly related to the total energy per particle of the system. In order to show that, we first notice that the potential energy per particle can be written in the form

$$\frac{\langle V \rangle}{N} = \frac{n}{2} \int d\mathbf{r} \, V_{dd}(\mathbf{r}) g(\mathbf{r}). \tag{8}$$

As shown in Fig. 3, the anisotropic contributions to the pair distribution function are much smaller than the isotropic mode, so we can approximate $g(\mathbf{r}) \simeq g_0(\mathbf{r})$. In dimensionless form, the potential energy per particle becomes

$$\frac{\langle \tilde{V} \rangle}{N} = x \pi e^{-2\gamma} \int d\tilde{r} \frac{g_0(\tilde{r})}{\tilde{r}^2}, \tag{9}$$

where $\tilde{r}$ and $\langle \tilde{V} \rangle$ are expressed in units of $a_0(\alpha)$, $E_0(\alpha) = \hbar^2/(ma_0^2(\alpha))$, respectively. As a result, $\langle \tilde{V} \rangle / N$ depends only on the gas parameter $x$ and on an integral of $g_0(\tilde{r}) = g_0(\frac{r}{a_0(\alpha)})$, which is left almost unchanged for all tilting angles. One thus concludes that, within this approximation, universality in the pair distribution function induces universality in the potential

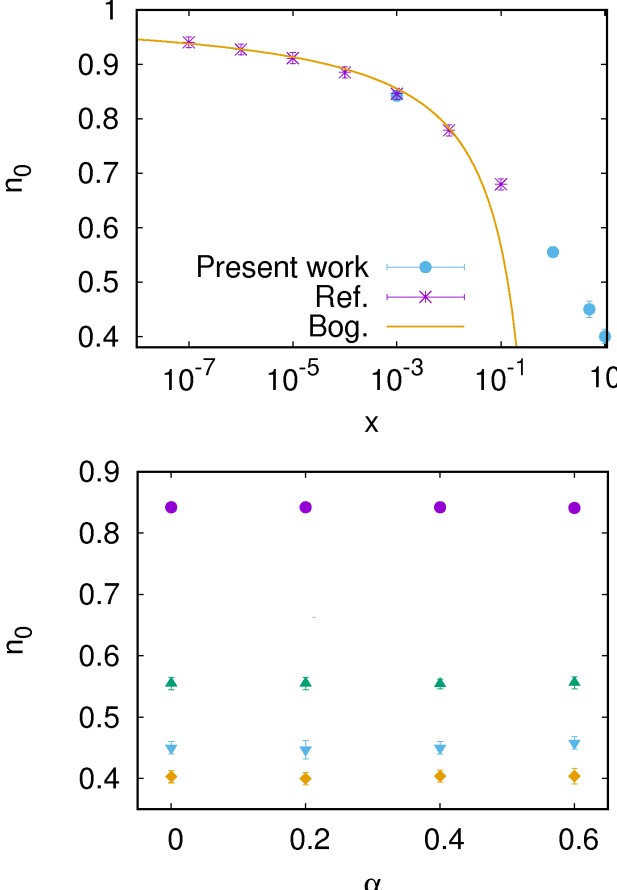

Figure 5: Condensate fraction as a function of the gas parameter (upper plot), and as a function of $\alpha$ for different gas parameters $x = 0.001$ (dots), 1 (up-triangles), 5 (down-triangles), and 10 (diamonds) (lower plot). In the upper plot, we provide the values obtained by Ref. [42], as well as the Bogoliubov prediction, which is plotted for $x < 1$.

energy per particle. This is in agreement with the universal properties of dipolar systems discussed in Ref. [40]. In order to link this result with the universal behaviour of the total energy per particle, we use of the Hellmann-Feynman theorem, which states that, for a Hamiltonian of the form $\hat{H} = \hat{H}_0 + u\hat{H}_1$, the ground state energy can be written as

$$E = E_0 + \int_0^1 du \, \langle \Psi(u)| \frac{d\hat{H}}{du} |\Psi(u)\rangle \,, \tag{10}$$

where $E_0$ is the ground state energy of $\hat{H}_0$. We now take $\hat{H}_0$ and $\hat{H}_1$ to be the kinetic and (dipolar) potential terms of the Hamiltonian, respectively. The case $u = 1$ corresponds to the full Hamiltonian considered in this work. With this choice, $E_0 = 0$ as this corresponds to the ground state energy of a free gas of bosons at zero temperature. In this way one has

$$\frac{E}{N} = \frac{1}{N} \int_0^1 du \, \langle \Psi(u)| V |\Psi(u)\rangle = \int d\mathbf{r} \, V_{dd}(\mathbf{r}) g(\mathbf{r}, u), \tag{11}$$

or, in scattering length units

$$\frac{\tilde{E}}{N} = x\pi e^{-2\gamma} \int_0^1 du \int d\tilde{r} \, \frac{g_0(\tilde{r}, u)}{\tilde{r}^2} \,. \tag{12}$$

In this expression, $g(\mathbf{r}, u)$ stands for the pair distribution function corresponding to a Hamiltonian $\hat{H} = \hat{H}_0 + uV$, with $0 < u < 1$. Our previous analysis has shown that already for $u = 1$ the contribution of the isotropic term dominates the pair distribution function in the range of tilting angles and gas parameters considered. By reducing the strength of the dipolar interaction with $u < 1$, as given in Eq. (12), the impact of the anisotropic modes is reduced as the potential contribution to the total energy is less relevant the lower $u$ is. Therefore, Eq. (12) links the universality of the pair distribution function to the universality of the total energy per particle. It must be remarked, however, that the low contribution of the anisotropic modes of the pair distribution function, compared to the isotropic one, is key to ensure universality in the energy, since the anisotropic modes are not universal as seen in Fig. 3. We conclude that the lack of strong anisotropic contributions in the structure of the system, even at large gas parameters and tilting angles, leads to universality in the total energy per particle. We further extend over this argument below.

Next, we discuss the condensate fraction $n_0$ of the system. This quantity is obtained from the large-distance limit of the off-diagonal one body density matrix

$$\rho_1(\mathbf{r}) = N \frac{\int d\mathbf{r}_2 \cdots d\mathbf{r}_N \Psi^*(\mathbf{r}_1 + \mathbf{r}, \mathbf{r}_2, \cdots, \mathbf{r}_N) \Psi(\mathbf{r}_1, \mathbf{r}_2, \cdots, \mathbf{r}_N)}{\int d\mathbf{r}_1 \cdots d\mathbf{r}_N |\Psi(\mathbf{r}_1, \cdots, \mathbf{r}_N)|^2}, \tag{13}$$

as $n_0 = \rho_1(|\mathbf{r}| \to \infty)/n$. The upper plot of Fig. (5) shows $n_0$ for different values of the gas parameter $x$ and tilting angle $\alpha$. Remarkably, the condensate fraction remains essentially constant at fixed $x$. We find an almost perfect scaling behaviour up to the largest value $x = 10$ explored, which stays largely away from the diluteness regime. As expected, the value of $n_0$ decreases with increasing $x$, as the enhancement of quantum fluctuations at larger densities favors the depletion of the condensate. In order to discern whether the dependence of the condensate fraction of $x$ is universal or not, we also to compare our results to the Bogoliubov prediction $n_0^{\mathrm{B}} = 1 - 1/|\ln x|$. As it can be seen, the Bogoliubov prediction is recovered only in the weakly interacting regime corresponding to $x \lesssim 0.001$, while significant deviations appear as $x$ increases. Still and as mentioned above, the condensate fraction has the same value for fixed $x$ and different $\alpha$, thus showing a clear scaling behavior as the previous quantities analyzed. For large $x$ the DMC prediction is significantly larger than the values obtained in the Bogoliubov model, as expected.

To conclude the numerical analysis, and considering all simulations have been performed with a fixed number of particles, we analyze how finite size effects influence our results. We report in the top panel of Fig. 6 the ratio of the energy of the dipolar gas at fixed $N$ to the corresponding value extrapolated to the thermodynamic limit ($N \to \infty$), denoted by $E_{\mathrm{ext}}$, for $x = 100$ and $\alpha = 0$. We also show the fit from where the extrapolated value is obtained, which corresponds to a function of the type $f(N) = E_{\mathrm{ext}} + C/\sqrt{N}$. This is consistent with finite size calculations performed in Ref. [43] for a system of non-tilted dipoles in two dimensions. Larger tilting angles produce similar results. In much the same way and in order to characterize the influence of finite size effects in other static properties, we report on the bottom panel of the same figure the pair distribution function for $N = 100$ and $N = 300$, also at $x = 100$ and $\alpha = 0$. In all cases, the box size $L$ is chosen such that the density is kept constant while changing $N$. As can be seen, finite size corrections for both quantities are small and below 0.75% already at $N = 100$, thus confirming that the universality conditions described above hold also in the thermodynamic limit.

After analyzing universality in the energetic and structural properties of the purely dipolar system, it is interesting to discuss what the effect of adding a short range isotropic interaction is. This is a relevant issue, as in actual experiments on dipolar systems this term is usually

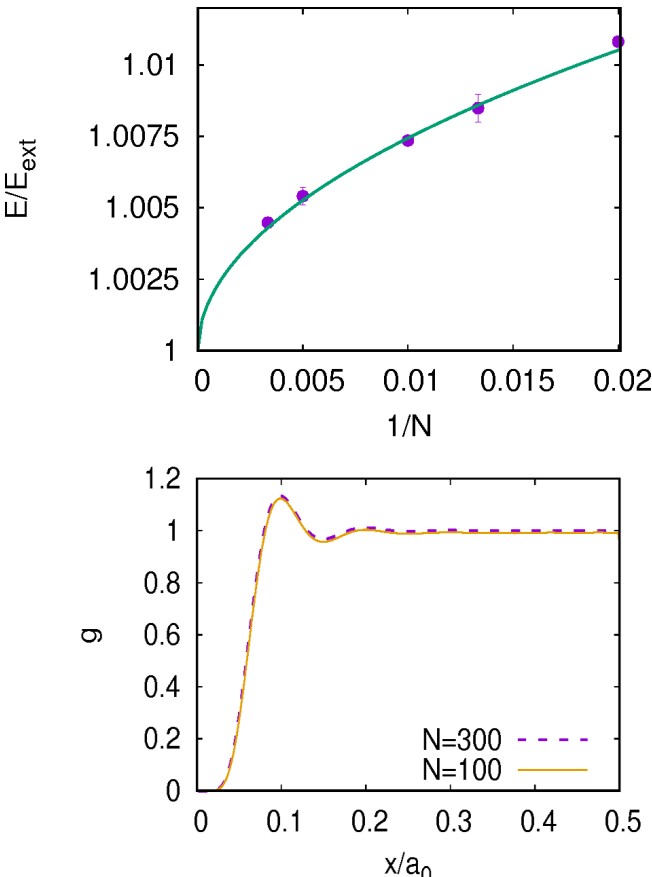

Figure 6: (Top) ratio of the DMC energies to the extrapolated $N \to \infty$ value for $x = 100$ and $\alpha = 0$. The solid line corresponds to a fit of the form $f(N) = E_{\text{ext}} + C/\sqrt{N}$ from where the extrapolated value is obtained. (Bottom) Pair distribution function at $x = 100$ and $\alpha = 0$ for $N = 100$ and $N = 300$ particles at the same density.

present. When it is included, the Hamiltonian becomes

$$H = -\frac{\hbar^2}{2m} \sum_{j=1}^{N} \nabla_j^2 + \sum_{i<j} V_{dd}(\mathbf{r}_{ij}) + V_{sr}(\mathbf{r}_{ij}) = -\frac{\hbar^2}{2m} \sum_{j=1}^{N} \nabla_j^2 + \sum_{i<j} V_{full}(\mathbf{r}_{ij}), \qquad (14)$$

where $V_{full}(\mathbf{r})$ stands for the total potential acting on the atoms. In order to explore how the inclusion of this short range term affects universality, we have chosen a model interaction of the form $V_{sr}(r_{ij}) = (\sigma/r_{ij})^6$. At this point, we have calculated the scattering length of the Hamiltonian in Eq. (14) from the large distance asymptotic behavior of the zero-energy solution of the two-body problem as discussed in [44]. This treatment is equivalent to solving the Scattering T-matrix of the system, and taking its zero-momentum limit. The result is shown in Fig. 7. As it can be seen, although non-zero, the addition of the short-range term, for moderate values of $\sigma$, does not alter significantly the scattering properties with respect to the purely dipolar model. In order to confirm that in the many-body case, we have performed additional simulations corresponding to the Hamiltonian in Eq. (14) with the chosen $V_{sr}(r)$. We have set $\sigma = 0.25$ in dipolar units, as in this case the presence of $V_{sr}(\mathbf{r})$ does not modify appreciably the total scattering length. We have computed the energy per particle and the pair distribution function at $x = 100$, and the condensate fraction at $x = 5$. The results are shown in Fig. 8, where we report, for $x = 100$ and different values of the tilting angle, the

ratio $E/E(\alpha = 0)$ (as in Fig. 1), the $g_0(r)$ and $g_2(r)$ modes of the pair distribution function (as in Fig 3), and the condensate fraction (as in the lower panel of Fig 5) for $x = 5$. As we can see from the figure, universality still holds in the main static properties of the system. The energy per particle departs only a few percent from the perfect universal behaviour (corresponding to $E/E(\alpha = 0) = 1$), and only at the largest polarization becomes slightly noticeable. The pair distribution function keeps being dominated by the isotropic mode, which clearly shows universality. In much the same way, the condensate fraction does not present a noticeable dependence on the tilting angle. We thus conclude that the presence of a short range repulsive potential does not alter significantly the universality present in the 2D dipolar gas when the DDI interaction is considered alone.

In view of the results reported in this work, we conclude that the universality displayed by the 2D dipolar gas is a direct consequence of the fact that the gas parameter must be increased to very large values ($x \simeq 400$) for the anisotropy of the DDI to have a strong influence in the structural properties of the system. This is furtherly supported by the fact that one can recover the properties reported in this work to a high degree of accuracy by replacing the full DDI by an isotropic potential of the form

$$V_{eff}(r, \alpha) = \frac{C_{dd}}{4\pi} \frac{\left(1 - \frac{3}{2}\sin^2 \alpha\right)}{r^3},\tag{15}$$

where the tilting angle becomes simply a parameter that tunes the effective strength of the interaction. This particular form of the potential makes its scattering length be almost identical to that of the fully anisotropic DDI interaction [39]. When the many-body Schrödinger equation for $V_{eff}(r)$ is expressed in scattering length units, it becomes independent of the polarization angle $\alpha$

$$-\frac{1}{2}\sum_{i=1}^{N}\tilde{\nabla}_i^2\Psi + \sum_{i<j}\frac{e^{-2\gamma}}{\left|\tilde{\mathbf{r}}_{ij}\right|^3}\Psi = \tilde{E}\Psi,\tag{16}$$

meaning that, in these units, solving Eq. (16) yields $\alpha$-independent results. Furthermore, since $V_{eff}(r, \alpha = 0) = V_{dd}(r, \alpha = 0)$, the properties obtained when solving Eq. (16) are the same as those obtained when solving the Schrödinger equation for the full $V_{dd}(r, \alpha = 0)$, written in

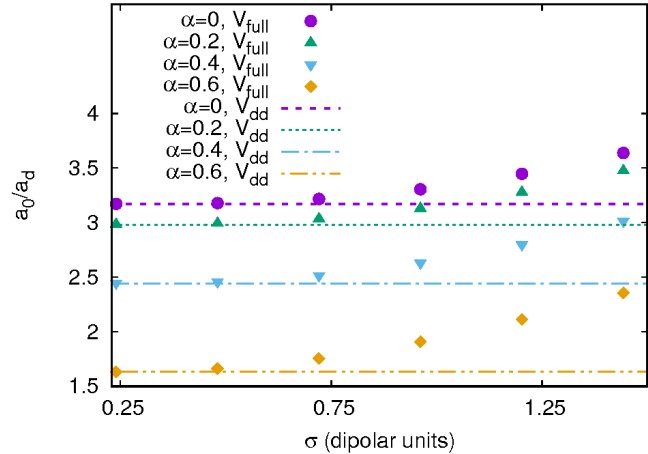

Figure 7: Scattering length of the compound system described by the Hamiltonian in Eq. (14) corresponding to a dipolar system with an additional Van der Waals tail, as a function of $\sigma$ and for different polarization angles. Here, $\sigma$ is given in dipolar units, with the characteristic length and energy scales given by $l = a_d$, $\epsilon_l = \hbar^2/ma_d^2$ respectively.

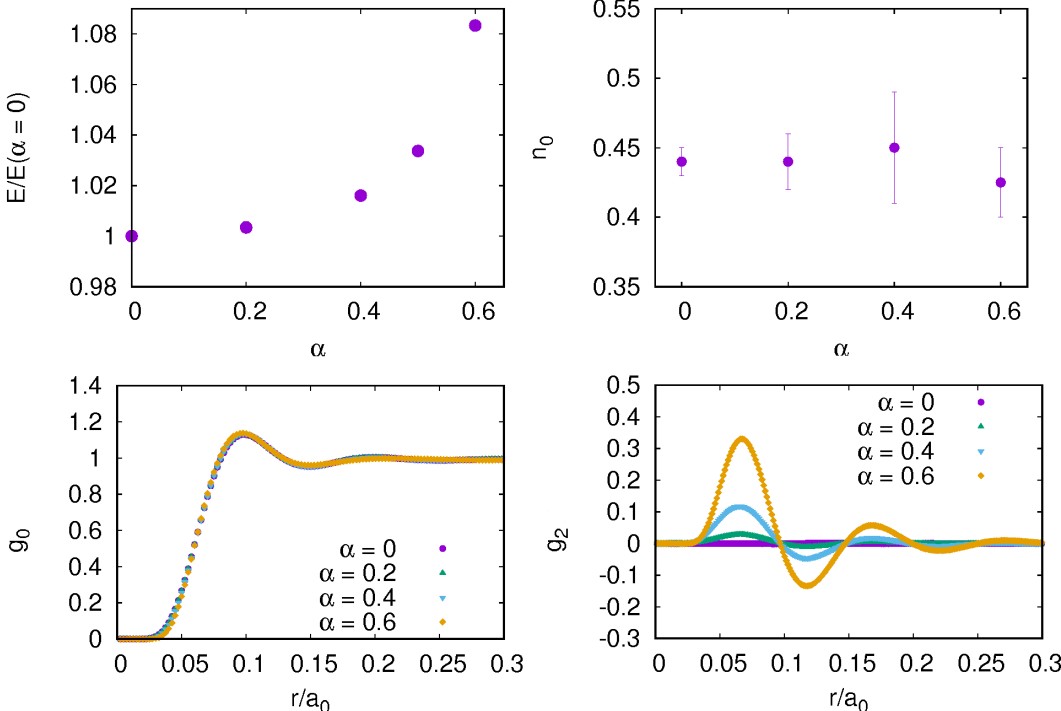

Figure 8: Ratio of the bulk DMC energy per particle, in units of the scattering length $a_0(\alpha)$ to $E(\alpha = 0, x)$, for $x = 100$ (top left panel). Condensate fraction as a function of the tilting values $\alpha$ for $x = 5$ (top right panel). Isotropic (bottom left panel) and first anisotropic modes (bottom right panel) of the pair distribution function for $x = 100$. All quantities have been computed for the Hamiltonian of Eq. 14, where a short range repulsive potential is considered along the DDI.

scattering length units. Thus, one can see from the results in Figs. 1, 3 and 5 that the energy per particle, the radial distribution function, and the condensate fraction of the system obtained when the potential $V_{eff}(r, \alpha)$ is considered (which correspond to the data for $\alpha = 0$ in the Figures) approximate reasonably well the results obtained with the full DDI when $\alpha \neq 0$. We believe that the fact that this isotropic approximation is successful, which is a consequence of universality, is due to the lack of a strong anisotropic influence in the structural properties of the system.

To summarize, we have studied the scaling of the dipolar interaction as a function of the polarization angle $\alpha$ and gas parameter $x$ in a system of two-dimensional bosonic dipoles. We have found that universality is lost already at $x \approx 0.001$ where the energy per particle deviates from the mean-field prediction as expected. Beyond that point, however, all energy curves collapse to a single one when properly scaled by the tilting-dependent scattering length $a_0(\alpha)$. This behavior holds up to surprisingly large values of $x$ close to the gas-stripe transition line, like $x = 350$, and up to large polarization angles near the collapse limit. In this same region, this scaling property is not only present in the energy, but also on the condensate fraction for all polarization angles considered ($\alpha \in [0, 0.6]$), and in the most relevant structural properties like the pair distribution function and the static structure factor. We have also shown that this behaviour is still present in the system when a short range repulsive potential is considered along with the DDI, which is typically the case in actual experiments. All this means that, for any $\alpha$ contained in the region considered, the angular dependence (and thus the anisotropic features) of the most relevant static properties of dipolar quantum Bose gases

in two dimensions are entirely contained in the $\alpha$-dependent $s$-wave scattering length, which is well approximated by the expression $a_0(\alpha)/a_d \approx e^{2\gamma}(1 - 3\lambda^2/2)$ with $\lambda = \sin(\alpha)$ and $a_d$ setting the dipolar length scale. From our analysis we finally conclude that the universal behavior of the dipolar Bose gas in two dimensions is a consequence of the overall low impact of the anisotropy on the structural properties of the system up to astonishingly large values of the gas parameter.

# Acknowledgments

The work has been supported by grant PID2020-113565GB-C21 from MCIN/AEI/10.13039/ 501100011033, and by the Danish National Research Foundation through the Center of Excellence "CCQ" (Grant agreement no.: DNRF156). J. Sánchez-Baena acknowledges funding by the European Union, the Spanish Ministry of Universities and the Recovery, Transformation and Resilience Plan through a grant from Universitat Politecnica de Catalunya.

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
