# Peer review of "Universal Properties of Anisotropic Dipolar Bosons in Two Dimensions"

_SciPost Physics, doi:SciPost Phys. 13, 031 (2022)_

## Round 1 · Referee Report · Anonymous (Referee 1) · 2022-2-4

Report

Dear Editors,

In the presented manuscript "Universal Properties of Anisotropic Bosons in Two Dimensions", the authors study numerically the effects of dipolar interactions in a purely two-dimensional gas of polarized bosonic dipolar particles as a function of the polarization angle and the gas parameter of the system. More specifically, the energy per particle, the pair distribution function, and the condensation fraction were considered. One of the main results is that, being properly rescaled, the energy per particle demonstrates a universal behavior up to very large values of the gas parameter when there are already strong deviations from the mean-field results.

I find the results of numerical calculation interesting and potentially useful, and I also do not see any reason to question their validity. There are, however, several issues concerning their understanding/interpretation and applicability to real physical systems, which need to be clarified and discussed:

  • The authors consider only the dipole-dipole interaction between bosons, completely ignoring any short-range contributions. This situation is, of course, experimentally possible via the Feshbach resonance, but very “singular”. In a typical situation, the short-range part of the interparticle interaction is present. How will the account of a, say, small short range interaction change the results?

  • Is the any connection between smallness of the anisotropic terms (at least for the values gas parameter smaller than 100) in the pair distribution function and the universal behavior of the energy per particle?

  • As a general comment, I find it strange that the authors do not present any argument which could give a hint to understanding of their numerical findings. Do the authors have some ideas why the found universal behavior of the energy per particles continues to such large values of the gas parameter – practically till the crystallization point around 400? In this respect, I would like to draw the author’s attention to the paper Physical Review Research 3, 013088 (2021) devoted to the exact relations for dipolar quantum gas, which could be potentially helpful.

To conclude, I do not recommend the manuscript in its present form for publication. In my opinion, there are several issues which has to be addressed before the final decision concerning publication can be made.

Yours sincerely,

---

## Round 1 · Referee Report · Anonymous (Referee 2) · 2022-3-1

Strengths

1) Extensive numerical analysis.

2) Clear presentation of the results.

3)Substantial novelty.

3) High impact topic.

Weaknesses

1) Lacking connection with existing literature.

2) Finite size scaling not reported.

Report

I read the paper with interest and I can certainly praise the neat presentation and the well delineated physical questions. The argument (universality in dipolar quantum gases) is highly relevant to AMO physics and this extensive numerical analysis will certainly attract the interest of many experts in the community.

The choice of presenting the numerical picture without pushing a definite theoretical interpretation helps to keep the article focused and improves its quality rather than reducing it. The observation of an extended universal scaling well beyond the expectations of local gases is certainly relevant and will surely produce several further theoretical investigations in the future.

Nevertheless, before the article is formally accepted the authors shall make a more substantial effort to connect their current picture with previous findings in the literature. Indeed, the small number of references in this paper is not justified, since the topic of universality in Bose gases has been extensively studied in the last decade.

The inadequate framing of the article in the literature is particularly evident in the discussion of universality for local interactions in d=2, where the authors refer to the book of Popov, Ref.[17], where ln|ln(x)| corrections to the mean-field universality of a weakly interacting Bose gas in 2D are discussed.

This picture is known to be incomplete, as the ln|ln(x)| terms are only observed in a negligible window of the interaction strength. The actual corrections are logarithmic ln|\mu x|, where \mu is a large experimental parameter \mu~380 according to numerical simulations see Refs. 1) https://journals.aps.org/prl/abstract/10.1103/PhysRevLett.87.270402 2) https://journals.aps.org/pra/abstract/10.1103/PhysRevA.66.043608

In particular, see the discussion below Eq. (3) in Ref. (1), which describes the impossibility to achieve the ln|ln(x)| correction regime. Notice that these studies have produced quite an impact in the community and several theoretical analysis have been devoted to reproduce them. See the RG studies: 3) https://doi.org/10.1103/PhysRevA.85.063607 4) https://doi.org/10.1103/PhysRevB.96.174505.

I encourage the authors to extend their discussion of the results in the light of the existing literature, providing a comparison of their findings with the current picture for local systems rather than with the textbook picture.

Finally, although the numerical analysis appears to be reliable, the paper does not contain any quantitative discussion of the finite size scaling for the system. The only sentence I could find on this aspect is: "We have checked that our results remain essentially unchanged when keeping n constant while increasing N and the box size L."

While I do not want to force the authors to perform an extensive numerical finite size scaling, it will be interesting to have at least a flash on how the convergence of the system to the thermodynamic limit is achieved.

I encourage the authors to extend their work and clarify those points.

Requested changes

1) Increase the discussion of current literature, in particular universality in local Bose gases.

2) Include at least a partial discussion of finite size scaling.

---

## Round 2 · Referee Report · Anonymous (Referee 1) · 2022-5-2

Report

Dear Editors,

I am fully satisfied with answers of the authors to my questions/comments and with changes made in the revised version of the manuscript "Universal Properties of Anisotropic Bosons in Two Dimensions". I therefore recommend it for publication because the manuscript discuss for the first time a very peculiar feature of a two-dimensional system of polarized dipolar particles – a universal behavior of various quantities that extends to surprisingly large values of the gas parameter.

Yours sincerely,

---

## Round 2 · Referee Report · Anonymous (Referee 2) · 2022-5-9

Report

Dear Editors,

I am quite satisfied with answers of the authors to my questions/comments and with the changes made in the revised version of the manuscript "Universal Properties of Anisotropic Bosons in Two Dimensions". The addition of Fig. 6 gives a good summary of the convergence of the thermodynamic quantities and justifies the choice of N=100 to perform the majority of the numerical
simulations.

The relation of the present findings to the existing literature of 2D quantum gases with contact interactions has also improved. However, the reference list is still incomplete for what concerns the experimental observation of quantum anomaly of 2D fermi gases. Indeed, evidences of quantum anomaly in the out-of-equilibrium dynamics of 2D fermi gases have been reported in reference:

"Quantum scale anomaly and spatial coherence in a 2D Fermi superfluid", DOI: https://doi.org/10.1126/science.aau4402

which should be acknowledged. Moreover, notice that Ref. [17] contains a misprint.

Once these minor changes have been implemented the paper can be published.

Yours sincerely,

Requested changes

Include Reference [P. A. Murthy, N. Defenu, L. Bayha, M. Holten, P. M. Preiss, T. Enss, S. Jochim, Science 365, 268-272 (2019)] in the discussion of experimental observation of the quantum anomaly in Fermi gases.

---

## Round 2 · Author Response

Reply to the Referee Reports

Report 1

Dear Editors,

In the presented manuscript "Universal Properties of Anisotropic Bosons in Two Dimensions", the authors study numerically the effects of dipolar interactions in a purely two-dimensional gas of polarized bosonic dipolar particles as a function of the polarization angle and the gas parameter of the system. More specifically, the energy per particle, the pair distribution function, and the condensation fraction were considered. One of the main results is that, being properly rescaled, the energy per particle demonstrates a universal behavior up to very large values of the gas parameter when there are already strong deviations from the mean-field results.

First of all, we would like to thank the Referee for his/her useful comments and suggestions, which have truly helped us to improve our work. In fact, we have incorporated them into the manuscript, providing new explanations that will help the reader to better understand the physics governing the discussed system.

In the following we elaborate on the points raised by the Referee. We believe that the responses given answer most (if not all) the questions, and hope that it provides sufficient ground for a publication in the SciPost Physics journal.

I find the results of numerical calculation interesting and potentially useful, and I also do not see any reason to question their validity. There are, however, several issues concerning their understanding/interpretation and applicability to real physical systems, which need to be clarified and discussed: - The authors consider only the dipole-dipole interaction between bosons, completely ignoring any short-range contributions.This situation is, of course, experimentally possible via the Feshbach resonance, but very “singular”. In a typical situation, the short-range part of the interparticle interaction is present. How will the account of a, say, small short range interaction change the results? Answer:

We agree with the Referee that, in typical experiments with Bose condensates, there is always a short-range component within the interatomic interaction. Its contribution can be made vanishing by Feshbach resonance technique (as actually it was done in pioneering works on dipolar condensates as for instance in Lahaye et al., Nature 448, 672 (2006)) leaving the dipolar interaction clean from other contributions. While, in a more general case there will be effects of the short-range part on the quantum properties of a system, those however are expected to be of less relevance in a dipolar system because of the quasi-long range character of the dipolar interaction in two dimensions. In order to arrive at this conclusion we have taken two actions.

(i) We have added a repulsive Van der Waals tail of the $V(r)=(\sigma/r)^6$ to the dipolar potential in dimensionless form. We have evaluated the scattering length of the compound interaction potential from the asymptotic behavior of the zero-energy wave function of the corresponding two-body problem, as described in N.N.Khuri et al., J.Math. Phys. 50, 072105 (2009), as a function of $\sigma$. This procedure is equivalent to a calculation of the zero-momentum limit of the scattering T-matrix. We have added a figure comparing the result to the value of the scattering length of the pure dipolar system, showing that the short-range part of $V(r)$ does not affect the result significantly.

(ii) Furthermore and as suggested by the Referee, we have performed additional many-body Diffusion Monte Carlo calculations. We include the same short-range repulsion potential $V(r$) with $\sigma$=0.25 in dipolar units, which ensures that the range of the interaction is fairly small compared to the range of the dipolar term. The value $\sigma=0.25$ also leaves the two-body scattering length unchanged. Under these circumstances, we have found that the universal properties obtained in the absence of the short-range interaction still hold. We have added both a discussion and two figures in the main text explaining these results.

  • Is there any connection between smallness of the anisotropic terms (at least for the values gas parameter smaller than 100) in the pair distribution function and the universal behavior of the energy per particle?

We thank the Referee for raising this question, which has helped us a lot to (hopefully) improve the contents of the Manuscript. There is, indeed, a direct connection between the pair distribution function and the energy. This can be seen in the following way: it is possible to write the potential energy per particle in scattering length units, as an expression that depends only on the gas parameter $x=n a_0^2$, with $n$ the density and $a_0$ the scattering length, and an integral of $g(r)$, with $g(r)$ the pair distribution function. From Fig. 3 of the main text, we know that $g(r)$ is dominated by the isotropic mode $g_0$, and that this isotropic mode is, essentially, a function of $r/a_0$, therefore the integral of $g(r)$ is also universal. In other words, the universality present in the pair distribution function makes the potential energy per particle depend mostly on the gas parameter, thus becoming essentially universal. From there, the universal character of the total energy per particle follows by direct application of the Hellmann-Feynman Theorem. This theorem states that the total energy of a system at zero temperature can be written in the form

$E = E_0 + \int_0^1 du <\Psi(u)|dH/du|\Psi(u)>$

where $|\Psi(u)>$ is the ground state wave function of a Hamiltonian $H = H_0 +u H_1$ with $u$ in (0,1). In this expression, $E_0$ stands for the ground state energy corresponding to $H_0$ (or $u=0$). Now we can take $H_0$ to be the kinetic operator and $H_1$ the potential term. In this case, $E_0 = 0$, since the ground state energy of a homogeneous gas of free bosons at $T=0$ is zero. This leads to

$E = \int du <\Psi(u)|V_{dd}|\Psi(u)>$

In this way, the total energy of the system can be expressed as the integral of the potential energy evaluated with the wave function $|\Psi(u)>$. Again, we can rewrite this equation as an expression that, in scattering length units, depends only on the gas parameter and on the integral of the pair distribution function g(r,u), which corresponds to the ground state of the Hamiltonian $H = H_0 + u H_1$. Since we know that $g(r)$ is mostly universal for $u=1$, it is expected it will also be universal if $0<u<1$, since decreasing u will make its on-universal, anisotropic components get reduced. Thus, we can see that the universality in the pair distribution function directly relates to the universality present in the total energy per particle, as the Referee suggested. We have included this discussion in the main text.

  • As a general comment, I find it strange that the authors do not present any argument which could give a hint to understanding of their numerical findings. Do the authors have some ideas why the found universal behavior of the energy per particle continues to such large values of the gas parameter – practically till the crystallization point around 400? In this respect, I would like to draw the author’s attention to the paper Physical Review Research 3, 013088 (2021) devoted to the exact relations for dipolar quantum gas, which could be potentially helpful.

This is also a very interesting question, as the universality goes much beyond the typical values one would expect. We believe that this is directly linked to the fact that the gas parameter of the system must be increased up to extremely large values ($x \simeq 400$) to see clear effects of the anisotropy in its structure. In other words, we believe universality to be connected with the lack of dominance of the anisotropic components for the gas parameters considered in our work. This is because we can mostly reproduce all the properties reported in our work by replacing the full DDI with an effective isotropic potential of the form

$V_{eff} = (C_{dd}/(4 \pi))(1-(3/2) \sin^2 \alpha)/r^3$ ,

which produces essentially the same scattering length as the fully anisotropic DDI interaction as discussed in Macia et al. Phys. Rev. A84, 033625 (2011). Since $V_{eff}$ coincides with the DDI at $\alpha=0$, in units of its s-wave scattering length $a_0$, where the characteristic length and energy scales are given by $a_0$, $E_{ff}= \hbar^2/m a_0^2$, this potential yields the same results that $V_{dd}(\alpha = 0)$ in units of the scattering length, and thus approximates very well all the properties of the system for different tilting angles. The fact that such an approximation is successful, which is a consequence of universality, is due to the lack of dominance of the anisotropy in the many-body system.

A complementary way to show that the influence of the anisotropy is a minor one is to consider once again the Hellmann-Feynman theorem. As stated in point 2 of this response, the total energy of the system can be written as an integral over the potential energies of systems with a rescaled DDI, with the potential energy per particle in scattering length units being essentially a function of the gas parameter and the integral of the pair distribution function. In this sense, universality in the total energy follows from universality in the pair distribution function (as we show in Fig. 3 in the main text). Since the anisotropic modes are clearly non-universal, if they were to dominate $g(r)$, no universality would be found in the energy. In this sense, again, the lack of strong effects by the anisotropy in the structure of the system in the gas parameter values considered in this work is what generates the observed universal behavior. We have introduced a paragraph before the conclusions discussing these ideas.

To conclude, I do not recommend the manuscript in its present form for publication. In my opinion, there are several issues which has to be addressed before the final decision concerning publication can be made.

We hope the answers provided address all the concerns of the Referee and leave the manuscript in a suitable form for publication in the SciPost Physics journal.

Report 2

I read the paper with interest and I can certainly praise the neat presentation and the well delineated physical questions. The argument (universality in dipolar quantum gases) is highly relevant to AMO physics and this extensive numerical analysis will certainly attract the interest of many experts in the community.

The choice of presenting the numerical picture without pushing a definite theoretical interpretation helps to keep the article focused and improves its quality rather than reducing it. The observation of an extended universal scaling well beyond the expectations of local gases is certainly relevant and will surely produce several further theoretical investigations in the future.

Nevertheless, before the article is formally accepted the authors shall make a more substantial effort to connect their current picture with previous findings in the literature. Indeed, the small number of references in this paper is not justified, since the topic of universality in Bose gases has been extensively studied in the last decade.

The inadequate framing of the article in the literature is particularly evident in the discussion of universality for local interactions in d=2, where the authors refer to the book of Popov, Ref.[17], where ln|ln(x)| corrections to the mean-field universality of a weakly interacting Bose gas in 2D are discussed.

This picture is known to be incomplete, as the ln|ln(x)| terms are only observed in a negligible window of the interaction strength. The actual corrections are logarithmic ln|\mu x|, where \mu is a large experimental parameter \mu~380 according to numerical simulations see Refs.

1) https://journals.aps.org/prl/abstract/10.1103/PhysRevLett.87.270402

2) https://journals.aps.org/pra/abstract/10.1103/PhysRevA.66.043608

In particular, see the discussion below Eq. (3) in Ref. (1), which describes the impossibility to achieve the ln|ln(x)| correction regime. Notice that these studies have produced quite an impact in the community and several theoretical analysis have been devoted to reproduce them. See the RG studies:

3) https://doi.org/10.1103/PhysRevA.85.063607

4) https://doi.org/10.1103/PhysRevB.96.174505.

I encourage the authors to extend their discussion of the results in the light of the existing literature, providing a comparison of their findings with the current picture for local systems rather than with the textbook picture.

We thank the Referee for suggesting useful references. Indeed, the overview of the literature for short-range interactions was quite limited in our original manuscript and also the explanation of the unusual structure of the expansion, in terms of the logarithm of the density, was missing. One reason why it is notoriously difficult to develop a precise perturbative description in two dimensions is the extremely weak dependence of the coupling constant $g$ on the s-wave scattering length $a$, which is of logarithmic form, i.e. $g=4\hbar^2/(m|\ln x|)$. This effect is specific to two dimensions and is also known as a quantum anomaly. Consequently, the expansion comes in terms of $\ln(x)$, $\ln|\ln(x)|$,... which requires to have exponentially small values of the gas parameter in order to see the $\ln|\ln(x)|$ correction regime. Indeed, in the Monte Carlo calculation of the coefficient in front of the double logarithm term, it was necessary to go to values of the gas parameter as low as $x = 10^{-100}$ [see G. E. Astrakharchik, J. Boronat, J. Casulleras, I. L. Kurbakov, and Yu. E. Lozovik Phys. Rev. A79, 051602(R) (2009).]. Nevertheless, such minuscule values are actually quite realistic due to an exponential dependence of the s-wave scattering length a_{2D} on the three-dimensional parameters. In the new version we provide additional information on the 2D physics and make a more complete overview of the existing literature.

Finally, although the numerical analysis appears to be reliable, the paper does not contain any quantitative discussion of the finite size scaling for the system. The only sentence I could find on this aspect is: "We have checked that our results remain essentially unchanged when keeping n constant while increasing N and the box size L."

Following the suggestion of the Referee, we have incorporated a figure showing quantitatively the finite size scaling of the system for zero tilting at x=100, as well as a paragraph discussing the results. Although not reported explicitly, a similar behavior is found for larger tilting angles. The figure shows that the choice of N=100 to perform the majority of our simulations is justified, since the energy per particle departs approximately only a 0.75% from the extrapolated value. On top of that, the pair distribution function remains essentially unchanged when comparing the N=100 result to the one obtained for a system of N=300 particles with the same density.

While I do not want to force the authors to perform an extensive numerical finite size scaling, it will be interesting to have at least a flash on how the convergence of the system to the thermodynamic limit is achieved.

I encourage the authors to extend their work and clarify those points.

Requested changes

1) Increase the discussion of current literature, in particular universality in local Bose gases.

2) Include at least a partial discussion of finite size scaling.

We hope that these changes address the concerns of the Referee in a satisfactory way and leave the manuscript in a suitable form for publication in the SciPost Physics journal.

---

## Round 2 · List of Changes

Added two paragraphs in the introduction where relevant aspects of universal properties of quantum systems in 2D are discussed, with added references.

Included a discussion starting at the end of pag.5 about the universality of the results in terms of the Hellmann-Feynman theorem.

Added a discussion about the relevance of finite size effects in the simulations starting in pag.6.

Added new figure 6 where the dependence of the energy and pair distribution function on the number of particles is reported.

Starting in pag.7 of the new manuscript we have added a discussion on the universal properties of the system when a short-range two-body interaction is added.

Added new figure 7 where er show the scattering length of a system with the DDI and a short-range interaction vs the scattering length of the DDI alone.

Added new figure 8 comparing energies, pair distribution functions and condensate fractions to check for universal behavior with an added short-range interaction.

---

## Round 3 · Author Response

Dear Editor,

Please find attached the last version of our manuscript

'Universal Properties of Anisotropic Dipolar Bosons in Two Dimensions'
by J.Sanchez-Baena, L.A. Peña Ardila, G.E. Astrakharchik and F. Mazzanti

Which has been reviewed already, asking for 'minor changes'.
We hope this new version is now suitable for publication.

Yours sincerely,

J.Sanchez-Baena, L.A. Peña Ardila, G.E. Astrakharchik and F. Mazzanti

---

## Round 3 · List of Changes

Added a few sentences to the last paragraph of page 1, where we extend a bit the discussion on the Quantum Anomaly, providing new references including the one requested by the last referee.

---

## Editorial Decision

published